
# Intensive aerosol properties of Boreal and regional biomass burning aerosol at Mt. Bachelor Observatory: Larger and BC-dominant particles transported from Siberian wildfires

Nathaniel W. May[1], Noah Bernays[1], Ryan Farley[2], Qi Zhang[2], and Daniel A. Jaffe[1]

[1]School of Science, Technology, Engineering, and Mathematics, University of Washington Bothell, WA 98011, USA
[2]Department of Environmental Toxicology, University of California Davis, CA, 95616, USA

*Correspondence to:* Nathaniel W. May (natemay@uw.edu)

**Abstract**. We characterize the aerosol physical and optical properties of 13 transported biomass burning (BB) events. BB events included long-range influence from fires in Alaskan and Siberian Boreal Forests transported to MBO in the free troposphere (FT) over 8-14+ days and regional wildfires in Northern California and Southwestern Oregon transported to MBO in the boundary layer (BL) over 10 h to 3 days. Intensive aerosol optical properties and normalized enhancement ratios for BB events were derived from measured aerosol light scattering coefficients ($\sigma_{scat}$), aerosol light absorbing coefficients ($\sigma_{abs}$), fine particulate matter ($PM_1$), and carbon monoxide (CO) measurements made from July to September 2019, with particle size distribution collected from August to September. The observations showed that the Siberian BB events had lower scattering Ångström exponent (SAE), higher mass scattering efficiency (MSE; $\Delta\sigma_{scat}/\Delta PM_1$), and a bimodal aerosol size distribution with a higher geometric mean diameter ($D_g$). We hypothesize that the larger particles and associated scattering properties were due to the transport of fine dust alongside smoke, in addition to contributions from condensation of secondary aerosol, coagulation of smaller particles, and aqueous phase processing during transport. Alaskan and Siberian Boreal Forest BB plumes were transported long distances in the FT and characterized by lower absorption Ångström exponent (AAE) values indicative of black carbon (BC) dominance in the radiative budget. Significantly elevated AAE values were only observed for BB events with <1 day transport, which suggests strong production of brown carbon (BrC) in these plumes but limited radiative forcing impacts outside of the immediate region.



## 1. Introduction

Biomass burning (BB) is a major source of atmospheric aerosols (Bond et al., 2013; Andreae and Merlet, 2001) and significantly impacts public health and regional air quality up to thousands of kilometers from the source (Jaffe et al., 2020; Boucher et al., 2013). BB aerosols also impact global climate by scattering or absorbing solar radiation, acting as cloud condensation nuclei, and altering cloud albedo (Boucher et al., 2013; Spracklen et al., 2011; Pierce et al., 2007). However, there are large uncertainties in BB aerosol formation, evolution, and radiative properties that limit our understanding of their climate impacts (Bond et al., 2013; Boucher et al., 2013; Bellouin et al., 2020; Boucher et al., 2020). BB emissions and their associated impacts are likely to increase globally due to hot and dry conditions resulting from climate change, particularly in the Western USA (Westerling, 2016; Liu et al., 2014) and sub-Arctic Boreal Forests of North America and Russia (Flannigan et al., 2009; Stocks et al., 1998).

BB particles are predominantly organic carbon (OC) and black carbon (BC), with some inorganic material (Vakkari et al., 2014; Reid et al., 2005; Zhou et al., 2017). BC is the most significant contributor to the absorption properties of BB particles (Healy et al., 2015; Bond et al., 2013), but OC also contributes to BB particle absorption in the form of brown carbon (BrC) (Andreae and Gelencsér, 2006). BC absorbs nearly uniformly in the range of 0.4 to 1 μm, resulting in an Absorption Ångström exponent (AAE) of ~1, while BrC can be identified in BB by an AAE > 2 due to its preferential absorption at lower wavelengths of sunlight (Andreae and Gelencsér, 2006). Atmospheric BrC is produced by incomplete combustion (Kirchstetter and Thatcher, 2012; Desyaterik et al., 2013; Lack and Langridge, 2013; Mohr et al., 2013) and secondary formation (Li et al., 2020; Nguyen et al., 2013; Updyke et al., 2012; Laskin et al., 2015). Photobleaching reduces the absorption of BrC and therefore significantly impacts the global radiation budget, but there are large uncertainties in this process (Liu et al., 2020). In contrast, atmospheric BC is chemically inert and primarily produced from flaming combustion (Healy et al., 2015; Bond et al., 2013). In situ aerosol optical measurements are essential to improved modeling of the contributions of BrC and BC to BB aerosol absorption properties (Brown et al., 2021).

As plumes age, BB particles grow from their initial diameter (30–100 nm) (Hosseini et al., 2010; Levin et al., 2010) and undergo chemical and physical changes (Carrico et al., 2016; Reid et al., 2005; Vakkari et al., 2014). Changes in particle size are primarily the result of coagulation and the condensation of secondary organic aerosol (SOA) onto existing particles (Reid et al.,



2005). However, the condensation of SOA is counterbalanced by loss due to evaporation and oxidation of primary organic aerosol during plume dilution (Collier et al., 2016; Zhou et al., 2017; Garofalo et al., 2019; May et al., 2013). The net condensation/evaporation effect in BB plumes can lead to an increase in PM mass due to SOA production (Hobbs et al., 2003; Yokelson et al., 2009; Vakkari et al., 2014), while others have observed limited or no net mass increase (Akagi et al., 2012; Jolleys et al., 2015; Garofalo et al., 2019). Even when PM mass concentrations do not change with age, particle diameter can shift through coagulation and particle-vapor mass transfer (Kleinman et al. 2020). Mie theory predicts mass scattering efficiency (MSE; $\sigma_{scat}$/PM) will increase as the average particle diameter grows toward the measurement wavelength (e.g., 300-700 nm) (Seinfeld and Pandis, 2006). Therefore, understanding the balance of aerosol condensation, evaporation, and removal processes is critical to understanding the scattering properties of aged BB particles.

Airborne mineral dust particles, which can be uplifted into the free troposphere alongside smoke by intense fire-related winds (Wagner et al., 2018), may also impact the radiative forcing of BB emissions. Dust particles are typically larger in size than smoke particles and can produce a net positive or negative radiative forcing depending on surface properties, particle size distribution, and composition (Balkanski et al., 2007; Durant et al., 2009; Russell et al., 2002). However, the climate properties of dust mixed with smoke are so far poorly understood. Dust emissions from fires are most frequently observed in arid regions (Chalbot et al., 2013; Nisantzi et al., 2014; Li et al., 2021) and can transported over thousands of kilometers (Ansmann et al., 2009; Clements et al., 2008; Baars et al., 2011). Recent observations of lofted dust from fires in the coniferous forests of the western USA (Maudlin et al. 2015, Schlosser et al. 2017, Creamean et al., 2016) and Russia (Popovicheva et al., 2014) suggest fires in non-arid regions may also emit dust. Despite in situ evidence, fires are not considered as a source of airborne mineral dust in climate or aerosol models.

In this study, we describe an overview of the intensive optical properties and normalized enhancement ratios of submicron aerosols (PM$_1$) observed during the summer of 2019 at the Mt. Bachelor Observatory (MBO), a remote, high-altitude site in the pacific northwest US. 13 BB events were observed, including smoke from nine regional fires (Northern California and Oregon; transported <1–3 days) and four Boreal Forest fires (Alaska and Siberia; transported 8-10+ days). Aerosol optical and physical properties of these events were explored with respect to source location, emission characteristics, and transport time. The goals of this work are to elucidate how



source and transport characteristics of BB events influences their climate impact through aerosol size distributions and associated scattering properties, as well as contributions of BrC and BC to absorption properties.

## 2. Methods

### 2.1 Sampling site

Measurements were conducted at the MBO, located near the summit of Mt. Bachelor (43.981°N 121.691°W, 2764 m a.s.l) from 1 July to 10 September 2019. These observations were part of the larger Fire Influence on Regional to Global Environments and Air Quality (FIREX-AQ) experiment (Liao et al., 2021; Xu et al., 2021; Decker et al., 2021; Wiggins et al., 2021; Makar et al., 2021), which included extensive observations at MBO (Farley et al., 2022). Due to its remote location and limited anthropogenic influence, MBO is an ideal site for measurements of wildfire plumes ranging from locally emitted to long-range transport events (Laing et al., 2016; Baylon et al., 2017; Wigder et al., 2013). The atmospheric conditions during this study were typical for a clean background location as the $PM_1$ concentration were relatively low (avg. $\pm 1\sigma$) ($2.8 \pm 3.8$ µg $m^{-3}$), consistent with periods without wildfire influence observed at MBO in 2013 ($2.8 \pm 2.8$ µg $sm^{-3}$) (Zhou et al., 2019).

Boundary layer dynamics play an important role in the diurnal variation of aerosol composition at MBO. During daytime, upslope flow can mix in boundary layer (BL) air to the sampling site, while at night the site is influenced by free tropospheric (FT) air masses. These regimes can be differentiated based on ambient water vapor concentrations (WV). ΔWV is calculated as the difference between the WV (g $kg^{-1}$) at the time CO peaked during the event and the WV previously found to demarcate the BL and FT (July-August: 5.23 g $kg^{-1}$, September: 4.60 g $kg^{-1}$) (Wigder et al., 2013; Baylon et al., 2015; Zhang and Jaffe, 2017). Typically, local smoke plumes will be seen in upslope BL air with a positive ΔWV, whereas distant smoke (Alaska, Siberia) will be transported in the FT with a negative ΔWV.

### 2.2 CO, Aerosol, & Meteorology

Details of the CO, aerosol, and meteorology measurements at MBO employed in the current study have been previously described in detail (Laing et al., 2016, 2020; Baylon et al., 2017) and thus only briefly described here. Basic meteorology measurements include temperature,



humidity, and wind speed (Ambrose et al., 2011). CO measurements were made using a Picarro G2302 cavity ring-down spectrometer. Calibrations were performed every 8 h using three different

National Oceanographic and Atmospheric Administration (NOAA) calibration gas standards, which are referenced to the World Meteorological Organization's (WMO) mole fraction calibration scale (Gratz et al., 2015). Total CO uncertainty based on the precision of calibrations over the campaign was 3%.

Dry (relative humidity (RH) less than 35 %) aerosol scattering and absorption coefficients,

aerosol number size distribution (30 – 600 nm), and particle mass were measured during the 2019 summer campaign in 5 min averages. An inline 1 µm impactor was located upline of the aerosol instruments. All particle measurements were corrected to standard temperature and pressure (STP; T = 273.15 K, P = 101.325 kPa) and reported this way throughout the manuscript. Aerosol light scattering coefficients ($\sigma_{scat}$) were measured by an integrating nephelometer (model 3563, TSI Inc.,

Shoreview, MN) at 450 (blue), 550 (green), and 700 (red) nm. Data reduction and uncertainty analysis for the scattering data are outlined by Anderson & Ogren (1998).

We measured aerosol light absorption coefficients ($\sigma_{abs}$) with a 3λ tricolor absorption photometer (TAP, Brechtel Inc., Hayward, CA) at wavelengths 467 (blue), 528 (green), and 660 (red) nm (Laing et al., 2020). Unless otherwise stated, $\sigma_{scat}$ and $\sigma_{abs}$ values represent measurements

taken at 550 and 528 nm, respectively. The absorption coefficients were corrected using the filter loading and aerosol scattering correction factors derived by Virkkula (2010). Uncertainty calculations were based on those used in a previous study at MBO for measurements with a 3λ PSAP (Fischer et al., 2010). Combining sources of uncertainties (Anderson et al., 1999; Bond et al., 1999; Virkkula et al., 2005) yielded total uncertainties for $\sigma_{abs}$ of 30–40 % during BB events.

Combined scattering uncertainties yielded lower total uncertainties for $\sigma_{scat}$ of 15-20% during BB events. However, the relative uncertainty between events for absorption and scattering are much lower (<10%). This was estimated by calculating the relative standard deviation of the absorption and scattering measurements over a 60 min time window with no wildfire influence at MBO.

The power law relationship between scattering and wavelength was used to adjust the 550

nm $\sigma_{scat}$ measurement to 528 nm using Eq. (1):

$$\sigma_{scat}^{528} = \sigma_{scat}^{550} \times \left(\frac{\lambda_{550}}{\lambda_{528}}\right)^{SAE_{450,550}}$$

(1)

where λ is wavelength and SAE is the scattering Ångström exponent calculated with the 450–550 nm pair. Absorption Ångström exponent (AAE) values were calculated for the $\sigma_{abs}$ pair of 467 and 660 nm using Eq. (2):

$$AAE = -\frac{\log\left(\frac{\sigma_{abs}^{467}}{\sigma_{abs}^{660}}\right)}{\log\left(\frac{467}{660}\right)}$$

(2)

Uncertainties for the intensive aerosol optical properties SAE and AAE values **(Table S1)** were calculated by propagating the uncertainties from the measurements used in the respective calculations using addition in quadrature (Fischer et al., 2010).

We measured 5 min averaged dry aerosol number size distribution with a TSI 3938 SMPS. The SMPS system consisted of a TSI 3082 electrostatic classifier with a TSI 3081 differential mobility analyzer (DMA) and a TSI 3787 water-based condensation particle counter. Geometric mean diameter ($D_g$) was calculated from lognormal fits of event averaged size distributions performed with the standard fitting algorithm of Igor Pro analysis software [fit parameters: $x_0 = D_g$, width = $2.303 \cdot 2 \log \sigma_g$, A = $2.303 \cdot N / (\pi \cdot width)$].

Dry particle mass under 1 µm ($PM_1$) was measured with an optical particle counter (OPC, model 1.109, Grimm Technologies, Douglasville, GA). This is a USA EPA equivalent method for measuring $PM_{2.5}$ mass concentration. August and September OPC data were determined to be artificially biased high, which coincided with the installation of a thermodenuder in the aerosol sampling line on July 31, 2019. Correction of Aug. – Sep. OPC $PM_1$ measurements using
coinciding high-resolution time-of-flight soot particle aerosol mass spectrometer (SP-AMS) $PM_1$ measurements (Farley et al., 2022) is presented in the **Supporting Information**.

### 2.3 Enhancement Ratio Calculation

As in prior studies of BB events at MBO (Laing et al., 2016; Briggs et al., 2016), normalized enhancement ratios (ΔY/ΔX, NER) of $\Delta\sigma_{scat}/\Delta CO$ and $\Delta\sigma_{abs}/\Delta CO$, and $\Delta PM_1/\Delta CO$
were calculated from the slope of the reduced major axis (RMA) regression of Y plotted against X. Intensive aerosol optical properties mass scattering and mass absorption efficiencies (MSE and MAE) were calculated as the NERs of $\Delta\sigma_{scat}/\Delta PM_1$ and $\Delta\sigma_{abs}/\Delta PM_1$, respectively, at 550 nm for $\sigma_{scat}$ and 528 nm for $\sigma_{abs}$. Single scattering albedo (ω) was calculated as the RMA regression of scattering and total extinction (scattering + absorption). In all cases the enhancements are large





compared to background, thus avoiding the problems described by Briggs et al. (2016) for small
enhancements above background. As in prior studies (Laing et al., 2016), uncertainties for the
NER calculations were determined from the uncertainties in the extensive properties used in
calculating the NERs and the uncertainty of the RMA regression using addition in quadrature.

Precision uncertainty and total uncertainty were calculated as described by Anderson et al.,
1999 for all values derived from optical measurements **(Table S1)**. Precision uncertainty is the
uncertainty associated with noise and instrument drift. Total uncertainty includes precision
uncertainty, the uncertainty associated with the corrections we applied to the data, and the
uncertainty associated with the calibration method. Precision uncertainty is best used to compare
the individual BB events seen at MBO in this study, whereas total uncertainty is more appropriate
to consider when comparing the measurements presented in this study with data collected using
other measurement methods.

### 3. Results & Discussion

### 3.1 BB Event Identification

The summer of 2019 was a relatively low activity fire season in the Pacific Northwest,
while higher temperatures in the Arctic and sub-Arctic contributed to extensive wildfires across
Boreal forests in Alaska and Siberia. We identified 13 BB events from Jul. – Sep. 2019, which
ranged from 2 to 44 h in duration (**Figure 1**). These were split when discernable plumes were
separated by a multi-hour drop below the BB event criteria of $\sigma_{scat} > 20$ Mm$^{-1}$ and CO > 110 ppbv.
Only 8% of the 1 h averages at MBO met our criteria for a BB event. Most observed BB events
exhibited average $\sigma_{scat}$ and CO in the ranges of 20-40 Mm$^{-1}$ and 130-170 ppbv, respectively. For
comparison, 51% of the 5 min averages met a higher BB event criterion ($\sigma_{scat} > 20$ Mm$^{-1}$ and CO
> 150 ppbv) during the high fire year of 2015, where most event average $\sigma_{scat}$ and CO ranged from
100-300 Mm$^{-1}$ and 200-400 ppbv, respectively (Laing et al., 2016). Continuing the approach from
(Laing et al., 2016), we use the term event, not plume, because of the long duration of some of the
events and the fact that some BB events observed in 2019 were influenced by emissions from
multiple fires. A comparison of CO, $\sigma_{scat}$, $\sigma_{abs}$, and PM$_1$ for BB event and background periods is
presented in **Figure S1**. BB events were characterized by a CO mean of 145 ppb  and median of
140 ppb, $\sigma_{scat}$ mean of 41.3 Mm$^{-1}$ and median of 33.6 Mm$^{-1}$, $\sigma_{abs}$ mean of 3.93 Mm$^{-1}$ and median



3.18 Mm$^{-1}$, and PM$_1$ mean of 10.8 µg m$^{-3}$ and median of 8.3 µg m$^{-3}$. During background periods,
we observed a substantially lesser values for CO (mean 103 ppb; median 104 ppb Mm$^{-1}$), σ$_{scat}$
(mean 7.4 Mm$^{-1}$; median 6.5 Mm$^{-1}$), σ$_{abs}$ (mean 0.53 Mm$^{-1}$; median 0.44 Mm$^{-1}$), and PM$_1$ (mean
4.8 µg m$^{-3}$; median 3.4 µg m$^{-3}$).

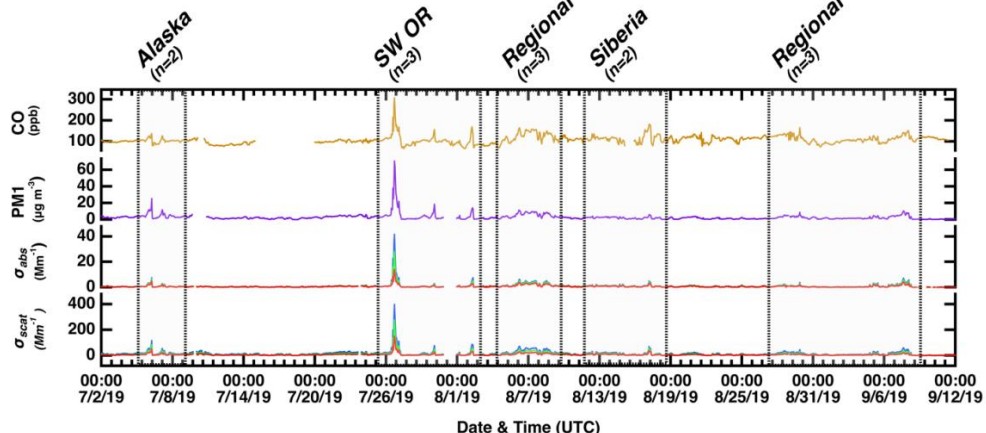

**Figure 1:** Time series (UTC) of the extensive properties CO (ppb), PM$_1$ (µg m$^{-3}$), scattering (Mm$^{-1}$
), and absorption (Mm$^{-1}$) measured at MBO from July 1, 2019, to September 14, 2019. BB event
periods are highlighted by grey fill dashed boxes, with source identification and number of discrete
BB events with σ$_{scat, green}$ > 20 Mm$^{-1}$ and CO > 110 ppbv per period above.

### 3.2 Source and Transport of BB Events

NOAA HYSPLIT airmass back trajectories and NASA MODIS Aqua and Terra images of
smoke were used to identify origin of each event and estimate transport time. Two BB events (7/5-
7/6), arriving at MBO with negative ΔWV (-1.53 g k$^{-1}$) **(Table 1)** indicative of FT transport, were
determined to originate from a complex of lightning-initiated fires in Boreal Forests of the interior
of Alaska (AK). The fires started from June 22 to July 2 and the smoke transported to MBO over
the course of 8-10 days **(Figures S2)**. Results from a FIREX-AQ model study by Makar et al.,
2021 previously suggested wildfires in AK impacted aerosol optical depth (AOD) in North
America during this period.

Three BB events with the shortest transport time to MBO observed in this study (~10-15
h) **(Figures S3)** originated from the Milepost 97 fire southeast of MBO near Canyonville, OR,
which was initiated on July 24$^{th}$ and burned 13,000 acres. Of the southwest Oregon (SW OR)
events, two exhibited positive ΔWV (0.15, 1.47 g k$^{-1}$) indicative of BL transport typically observed
for local fires, and one exhibited a negative ΔWV (-0.70 g k$^{-1}$) indicative of FT arrival **(Table 1)**.





Six BB events that had multiple fires active along the back trajectory were categorized as "Regional" in **Table 1**. The August regional events may have been influenced by emissions from forest fires in Washington, Oregon, and/or California, with transport times estimated to be 1-3 days **(Figure S4-S5)**. One August regional event had $\Delta$WV near zero, indicative of mixed BL and FT influence. The September regional BB events originated from multiple forest fires directly south of MBO in Oregon and California, with transport times estimated to be <12-48 h **(Figure S6)**. September regional events were characterized by a positive $\Delta$WV (0.85 - 4.2 g k$^{-1}$) consistent with prior observations of BL transport of regional BB events to MBO.

Fires in the Boreal Forest of Siberia near Lake Baikal were initiated by lightning in July 2019 and continued through August, burning over 3 million hectares with smoke transported eastward towards Alaska and the continental USA. 10-day air mass back trajectories from MBO for the peak 8/12 and 8/17 BB events passed over Alaska, but wildfire activity in Alaska had largely subsided. Coinciding measurements by Johnson et al. 2021 observed long-range transported Siberian BB emissions impacting western Canada, with significant aerosol layers between 3-10 km a.g.l. measured by LIDAR on 8/7, 8/10, and 8/13-14. Their analysis of MODIS observations and near-real-time satellite-based emissions from the Quick-Fire Emission Database demonstrated enhanced Siberian BB emissions between July 19 and August 14, 2019. In addition, NASA GEOS-CF global model simulations showed the emissions from 2019 Siberia fires were transported across the Pacific Ocean and Arctic region (Johnson et al., 2021). Therefore, Siberian Boreal Forest fires were determined to be the origin of 8/12 and 8/17 BB, with transport time estimated >14 days **(Figure S7)**. The negative $\Delta$WV of the Siberian events (-2.32, -0.73 g k$^{-1}$) were indicative of long-range transport to MBO occurring in the FT, which is consistent with established transport pathways of Siberian BB events (Laing et al. 2016).

**3.3 Overview of Intensive Aerosol Properties of BB aerosol at MBO**

**Table 1** provides an overview of normalized enhancement ratios (NERs) and intensive optical properties of 13 BB events of Siberian, Alaskan, SW OR, and regional origin observed at MBO during the summer of 2019. Observed BB event ranges of $\Delta\sigma_{scat}/\Delta$CO (0.30 to 1.29 Mm$^{-1}$ ppbv$^{-1}$), $\Delta\sigma_{abs}/\Delta$CO (0.01 to 0.13 Mm$^{-1}$ ppbv$^{-1}$), and $\Delta$PM$_1$/$\Delta$CO (0.04 to 0.37 µg cm$^{-3}$ ppbv$^{-1}$) were comparable to prior summer measurements of transported BB aerosol (Laing et al., 2016; Baylon et al., 2017; Wigder et al., 2013). The range of BB event $\omega$ ($\sigma_{scat}/(\sigma_{scat}+\sigma_{abs})$) observed in



summer 2019 (0.88 – 0.95) was lower, (i.e., more absorbing) than previously observed in summer 2015 at MBO (0.95 - 0.98). The observed range of event average AAE (0.97 to 2.55) was also lower, indicative of greater BC than previous observations in summer 2015 (2.3 to 4.12). The

265 precision and total uncertainties of intensive parameters derived from extensive measurements are provided for these events in **Table S1.**





**Table 1:** Overview of intensive plume properties observed June – September 2019 at MBO. ΔWV was calculated as the difference between the WV at the peak CO of the event and the cutoff value. AAE and SAE values are event averages of hourly data and all other normalized enhancement ratios (NER) were calculated as the slope of reduced major axis (RMA) regression of hourly data. MSE is the NER of $\Delta\sigma_{scat}/\Delta PM_1$ and MAE is the NER of $\Delta\sigma_{abs}/\Delta PM_1$. All were calculated from 550 nm (green) measurement wavelength data. Geometric diameter ($D_g$) was calculated from lognormal fits of event averaged SMPS size distributions. Category values at the bottom of the table are averages. Events with measurements reported by Farley at al. 2022 are identified by an asterisk.

| Event date & time (UTC) | Transport time (days) | Source fire | ΔWV (g kg⁻¹) | Δσscat/ΔCO (Mm⁻¹ ppbv⁻¹) | Δσabs/ΔCO (Mm⁻¹ ppbv⁻¹) | ΔPM₁/ΔCO (µg cm⁻³ µg⁻¹ cm⁻³) | MSE (Mm⁻¹ µg⁻¹ cm⁻³) | MAE (Mm⁻¹ µg⁻¹ cm⁻³) | AAE | SAE | ω | Dg (nm) |
|---|---|---|---|---|---|---|---|---|---|---|---|---|
| 7/5/19 20:00 – 7/6/19 7:00 | 8-10 | Alaska | -2.32 | 1.29 | 0.10 | 0.35 | 3.71 | 0.30 | 1.27 | 1.64 | 0.93 | -- |
| 7/7/19 2:00 – 7/7/19 5:00 | 8-10 | Alaska | -0.73 | 0.88 | 0.069 | 0.25 | 3.59 | 0.28 | 1.33 | 1.75 | 0.9 | -- |
| 7/26/19 10:00 – 7/27/19 6:00 | <1 | SW OR | 0.15 | 1.25 | 0.13 | 0.33 | 3.83 | 0.41 | 2.55 | 1.94 | 0.91 | -- |
| 7/30/19 0:00 – 7/30/19 3:00 | <1 | SW OR | -0.70 | 0.82 | 0.044 | 0.28 | 2.98 | 0.16 | 1.30 | 2.39 | 0.95 | -- |
| 8/2/19 4:00 – 8/2/19 8:00 | <1 | SW OR* | 1.47 | 0.50 | 0.049 | 0.15 | 3.38 | 0.33 | 2.33 | 2.17 | 0.93 | 156 |
| 8/5/19 23:00 – 8/7/19 19:00 | 1-3 | Regional* | -- | 0.54 | 0.061 | 0.12 | 4.51 | 0.51 | 1.34 | 1.91 | 0.90 | 151 |
| 8/7/19 22:00 – 8/8/19 1:00 | 1-3 | Regional* | -- | 0.42 | 0.042 | 0.09 | 4.63 | 0.47 | 0.97 | 1.87 | 0.91 | 156 |
| 8/8/19 6:00 – 8/8/19 20:00 | 1-3 | Regional* | -- | 0.60 | 0.075 | 0.14 | 4.24 | 0.53 | 1.34 | 1.98 | 0.89 | 158 |
| 8/12/19 9:00 – 8/12/19 11:00 | >14 | Siberia | -- | 0.29 | 0.019 | 0.04 | 7.24 | 0.48 | 1.09 | 1.24 | 0.94 | 48, 231 |
| 8/17/19 2:00 – 8/17/19 9:00 | >14 | Siberia* | -2.82 | 0.50 | 0.035 | 0.05 | 10.28 | 0.72 | 1.39 | 1.11 | 0.93 | 92, 278 |
| 8/28/19 13:00 – 8/28/19 16:00 | 1-2 | Regional* | 0.85 | 1.24 | 0.19 | 0.36 | 3.38 | 0.32 | 1.53 | 2.18 | 0.90 | 151 |
| 8/29/19 21:00 – 8/29/19 22:00 | 1-2 | Regional* | 4.20 | 0.39 | 0.042 | 0.12 | 3.35 | 0.36 | 1.60 | 2.31 | 0.89 | 147 |
| 9/7/19 10:00 – 9/8/19 3:00 | 1-2 | Regional* | 1.27 | 0.76 | 0.11 | 0.20 | 3.70 | 0.55 | 1.78 | 2.27 | 0.88 | 162 |
| Alaska (n=2) | | | -1.52 | 1.09 | 0.085 | 0.30 | 3.65 | 0.29 | 1.30 | 1.69 | 0.93 | -- |
| SW Oregon (n=3) | | | 0.31 | 0.86 | 0.074 | 0.25 | 3.40 | 0.30 | 2.06 | 2.17 | 0.93 | 156 |
| Siberia (n=2) | | | -2.82 | 0.40 | 0.027 | 0.045 | 8.76 | 0.60 | 1.24 | 1.17 | 0.94 | 70, 254 |
| Regional (n=6) | | | 1.27 | 0.66 | 0.087 | 0.17 | 3.97 | 0.46 | 1.43 | 2.09 | 0.89 | 154 |



### 3.4 Intensive Aerosol Properties of BB aerosol at MBO

We observed substantial differences in the intensive physical and optical aerosol properties of Siberian, Alaskan, SW OR, and regional BB events. The average ($\pm 1\sigma$) $\Delta$ PM$_1$/$\Delta$CO of SW OR ($0.25 \pm 0.09$ µg cm$^{-3}$ ppbv$^{-1}$) events transported <13 hrs and regional events ($0.17 \pm 0.10$ µg cm$^{-3}$ ppbv$^{-1}$) transported 1-3 days were comparable to the $\Delta$PM$_1$/$\Delta$CO emission factor ratio ($0.16 \pm 0.11$ µg cm$^{-3}$ ppbv$^{-1}$) for fresh BB in temperate regions (Wigder et al., 2013; Akagi et al., 2012). Therefore, they can be characterized as relatively balanced in PM production and loss. In contrast, Siberian events were characterized by the lowest average $\Delta$PM$_1$/$\Delta$CO ($0.045 \pm 0.007$ µg cm$^{-3}$ ppbv$^{-1}$). This result suggests that net PM$_1$ loss through deposition, evaporation and/or cloud processing was greater than secondary aerosol production during >14-day transport in the FT from Siberia. Concurrent SP-AMS measurements in summer 2019 demonstrated a clear decreasing trend of $\Delta$OA/$\Delta$CO with increased transport time ranging from a few hours to over 10 days, for which PM losses during transport was provided as a possible reason (Farley et al., 2022). Wigder et al., 2013 and Weiss-Penzias et al., 2006 previously reported a similar association of decreased $\Delta$PM$_1$/$\Delta$CO with increased transport >750 km for wildfire smoke arriving at MBO in 2004-2011. However, measurements of elevated $\Delta$PM$_1$/$\Delta$CO for the 8-10 day transported Alaska BB events ($0.30 \pm 0.07$ µg cm$^{-3}$ ppbv$^{-1}$) demonstrates long-range transport in the FT is not always associated with greater PM$_1$ loss. Similarly, there was no significant difference in the $\Delta$PM$_1$/$\Delta$CO of Summer 2015 Siberian and regional BB events, which Laing et al., 2016 attributed to dry FT transport with limited precipitation. Observations at MBO thus indicate that long-range transport in the FT can yield a wide range in BB event $\Delta$PM$_1$/$\Delta$CO depending on removal characteristics.

BB aerosol lofted to the FT are typically less likely to be removed by dry deposition and can have a longer atmospheric lifetime of up to 40 days, but wet deposition can enhance removal and therefore reduce lifetime (Bond et al., 2013). Analysis of the impact of wet deposition of Canadian BB plumes demonstrates that it is the dominant mechanism for BC removal from the atmosphere and consequently determines lifetime and atmospheric burden (Franklin et al., 2014 and Taylor et al., 2014). Notably the 2019 Siberian events were transported over the Arctic, where inefficient long-range transport in the summer is attributed to meteorological conditions enhancing wet deposition (Mori et al., 2020). Raut et al., 2017 determined a large fraction of BC particles in Siberian plumes transported to the Arctic region are mixed with sufficient water-soluble



compounds to become CCN active and are scavenged by large-scale precipitation and wet convective updrafts. They found the role of dry deposition to be minor and limited to the lower troposphere. We used precipitation rates from NOAA HYSPLIT ensemble back trajectories for the AK (**Figure S2,** 10 days back trajectory) and Siberian (**Figure S7**, 14 days back trajectory) BB events as a measure of wet deposition during long-range transport. Average precipitation rates for

Siberian events (0.0038 mm h$^{-1}$) were more than double Alaskan events (0.0016 mm h$^{-1}$) **(Figure S8)**. Increased wet deposition during trans-Arctic FT transport thus likely contributed to the reduced $\Delta PM_1/\Delta CO$ in Siberian BB events observed at MBO. Indeed, Farley et al., 2022 concluded that the larger accumulation mode (700 nm $D_{va}$) in the during Siberian BB event $PM_1$ mass size distribution was the result increased aqueous-phase cloud processing during transport.

Examination of intensive aerosol absorption properties suggests that the Siberian BB aerosol were more absorptive and BC dominant than other BB events observed. Siberian BB events exhibited a higher average MAE ($0.60 \pm 0.17$ m$^2$ g$^{-1}$) than the AK, SW OR, and regional events ($0.29 \pm 0.01$, $0.30 \pm 0.13$, $0.46 \pm 0.09$ m$^2$ g$^{-1}$). BC enhancement in Siberian events identified in concurrent summer 2019 SP-AMS measurements at MBO by Farley et al., 2022 likely contributed

to the observed elevated MAE. Further information on the relative contributions of BC and BrC to BB absorptivity to the transported BB events observed at MBO can be inferred from the intensive aerosol optical property AAE **(Figures 2 & 3)**. Consistent with SP-AMS measurements, the long-range transported Siberia BB events exhibited low AAE values ($1.24 \pm 0.21$) indicative of BC dominance. Alaska BB events exhibited similar low AAE values ($1.30 \pm 0.04$) indicative of

BC dominance but were not characterized by enhanced MAE. SP-AMS composition measurements were unavailable for the Alaskan BB event. Prior observation of Siberian events at MBO with similarly enhanced MAE and lower AAE were hypothesized to originate from hotter, more flaming portions of the fires (Laing et al., 2016). Flaming fires are characterized by enhanced BC emissions and pyro-convective energy needed to loft the plume high into the atmosphere where

it can undergo long range transport. ARCTAS-A aircraft measurements in Alaska of a much larger BC/CO ratio in Siberian fire plumes than North American fire plumes (Kondo et al., 2011) provides further evidence that the emissions from flaming portions compose the majority of BB transported from Siberian Boreal Forest fires to North America.

An alternative explanation for low AAE Siberian and Alaskan BB events is BrC removal

by photolysis and oxidation during week-long transport (Laing et al., 2016). Analysis of 10 years



of aerosol properties at MBO by (Zhang and Jaffe, 2017) found Asian long-range transport wildfires with lower AAE (1.45 ± 0.02 and 1.54 ± 0.39) than PNW BB events (1.81 ± 0.59). Similarly, we found a general decrease in AAE with increased transport time (**Figure 2)**. Regional events with 1-3 days transport had a slightly elevated AAE (1.43 ± 0.19) indicative of slightly

more BrC mixed with BC. The highest AAE (2.06 ± 0.67) indicative of strong BrC was observed for the short-range transported (10-15 h) SW OR events. The highest values of AAE for the SW OR event were associated with the highest $PM_1$ concentration (**Figure S9**), likely due to the presence of nearby, fresh, and concentrated plumes. These results are consistent with the short half-life (~9 h) of BrC and prior measurements of AAE declining as BrC decays over the course

of hours with little remaining after days of atmospheric transport (Forrister et al., 2015; Hems et al., 2021).

        The highest scattering ω (0.94 ± 0.01) was observed in Siberian plumes. The average ω for Alaska (0.93 ± 0.01) and SW OR (0.93 ± 0.01) events were elevated compared to the ω for regional events (0.90 ± 0.01). The relatively high ω for Siberian events is thus consistent with our finding

of significantly elevated MSE ($\Delta\sigma_{scat}/\Delta PM_1$) (8.76 ± 2.15 $m^2\ g^{-1}$). Satellite observations and 3D atmospheric modeling of 2016 Siberian BB suggested that an observed increase in ω with transport time was likely due to atmospheric processing of SOA producing an increase in the mass scattering efficiency of BB aerosol (Konovalov et al., 2021). However, the MSE of long-range transported Siberian smoke plumes arriving at MBO in 2015 exhibited no difference from regional events

(Laing et al., 2016) and MSE values of aged BB plumes are typically < 6 $m^2\ g^{-1}$ (Hand and Malm, 2007; McMeeking, 2005). No substantial differences were observed in the average MSE values for AK, SW OR, and regional events (3.65 ± 0.08, 3.40 ± 0.42, 3.97 ± 0.56 $m^2\ g^{-1}$). According to Mie Theory, the uniquely enhanced scattering efficiency of the Siberian BB events suggests an increase in particle size towards the measurement wavelength (Seinfeld and Pandis, 2006).

To better understand the enhanced scattering observed in Siberian events we related the variation in the MSE values that we observed with aerosol size distribution measured by SMPS. We found SMPS geometric mean diameter ($D_g$) to be correlated with MSE across all 2019 BB events ($R^2$ = 0.93) **(Figure 4)**, which is consistent with Mie theory and previous ambient observations of aged BB (Lowenthal and Kumar, 2004; Laing et al 2016) and laboratory studies

(McMeeking, 2005). However, MSE in 2019 was ~40% higher than 2015 at a similar $D_g$ (Laing et al., 2016). It was suggested that larger size distributions in 2015 BB events may be the result of



more concentrated BB plumes with slower particle evaporation rates, leading to greater net PM and SOA accumulation during aging (Hodshire et al., 2019). This conclusion was supported by the correlation of 2015 BB event $D_{pm}$ with CO, $\sigma_{scat,}$ and $PM_1$, which can be thought of as

surrogates for plume concentration. However, $D_{pm}$ was not found to be correlated with CO, $\sigma_{scat,}$ or $PM_1$ in 2019, with the largest particles observed in the more dilute Siberian event. Plume concentration does not appear to be the determining factor in the particle size of 2019 BB events.

SAE is an intensive aerosol optical property that can be used as measure of particle size with small values indicating large sizes (Kleinman et al., 2020). All SAE values here **(Figures 2**

**& 3)** are expected to be > 1 because only fine particles ($PM_1$) were measured. Aug.-Sep. BB event SAE were negatively correlated with $D_g$ ($R^2 = 0.85$) **(Figure 4)**, generally supporting its use as a measure of particle size. SW OR and regional events exhibited elevated SAE values characteristic of the smaller size distributions of fresh to moderately aged BB plumes (May et al., 2014; Levin et al., 2010). In contrast, both long-range transported AK and Siberia events exhibited lower SAE

values suggesting that long-range transport may produce a shift in BB aerosol to larger particle sizes (Jung et al., 2012). However, particle morphology, composition and coatings can also alter SAE values. Thin coatings on BC particles may yield lower SAE values (Zhang et al., 2020).

Multiple field observations have observed $D_g$ and MSE to increase as a function of photochemical age (e.g., Akagi et al., 2012; Carrico et al., 2016; Kleinman et al. 2020). An increase

in MSE with age suggests a rearrangement of particle mass that favors large diameter efficient scatters at the expense of small inefficient scatters. This could be the result of a transfer of mass between the gas and particulate phases (i.e., condensation) and amongst particles (i.e., coagulation) (Kleinman et al., 2020). We explore the relationship between SMPS number size distributions and SP-AMS aerosol composition measurements in **Section 3.5** to examine potential influence of

aerosol growth processes during transport of Siberian BB events.

While secondary aerosol growth and/or coagulation during transport present plausible explanations for larger particles, the relationship between SAE and AAE of Siberian BB events suggests that we must consider the unexpected contribution of dust mixed with smoke. Notably, a portion of the 8/17 Siberian event exhibited a combination of elevated AAE and low SAE **(Figure**

**3)** that is typically indicative of dust aerosols with enhanced absorption at short wavelengths (Bergstrom et al., 2007; Titos et al., 2017). SAE has previously been used by Zhang and Jaffe 2016 to identify larger particle sizes in long-range transport of industrial pollution from Asia, which was





suggested to be due to mixing with mineral dust sources. In addition, prior measurements in spring have identified an association of dust mixed with smoke in long-range transported plumes from

Asia with lower SAE at MBO (Fischer et al., 2010).

        Wagner et al., 2018 illustrated via high-resolution eddy simulation that the energy released by wildfires leads to a significant increase in near-surface wind speed and enhanced dust uplift potential. Based on this model, we hypothesize the enhanced pyro-convective energy of Siberian fires that injects BC into the FT (Laing et al., 2016) may also produce intense fire-related winds

that lofted fine dust alongside smoke into the FT where it was transported to MBO. Enhancements in much larger coarse mode particles are typically used to identify the influence of dust (Lee and Cho, 2007), but recent measurements in the western USA found evidence of fine mode dust mixed with smoke (Maudlin et al., 2015; Schlosser et al., 2017; Jahn et al., 2021). Therefore, we propose that dust mixed with smoke may have contributed to the enhanced larger particle size distribution

in the Siberian BB events. In the following sections we present satellite data showing the transport of dust alongside smoke from Siberian Boreal Forest fires (**Section 3.4**) and SMPS size distributions to examine the potential contributions of dust to Siberian BB events (**Section 3.5).**

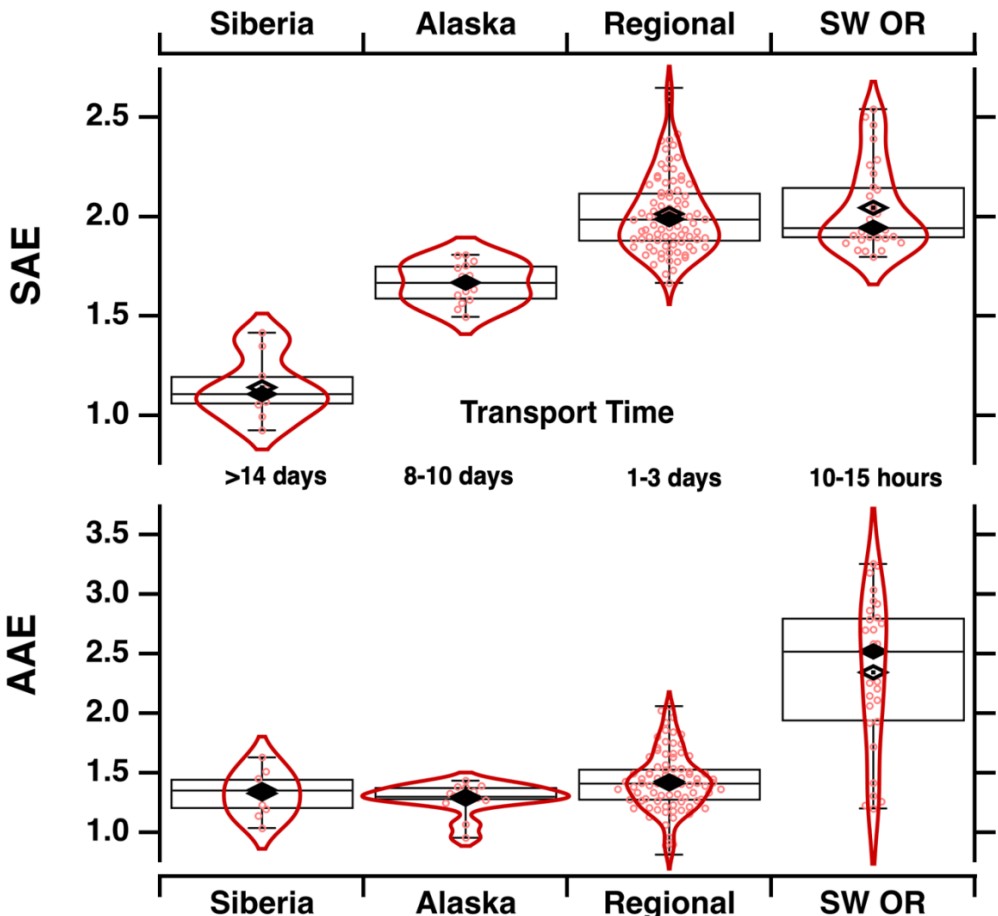

**Figure 2:** Combined violin and box plots of 1 hr average SAE and AAE values for July – September 2019 BB events observed at MBO grouped by source (Siberia, Alaska, Regional, SW OR). SAE describes particle size with small value indicating large size and AAE describes particle composition with larger values indicating greater BrC content. Violin plots in red show the rotated Gaussian kernel probability density, with mean shown as an open diamond, median shown as a solid diamond, and 1 hr average values shown as open circles. Boxes in black represent upper and lower quartiles, with whiskers representing the minimum and maximum.





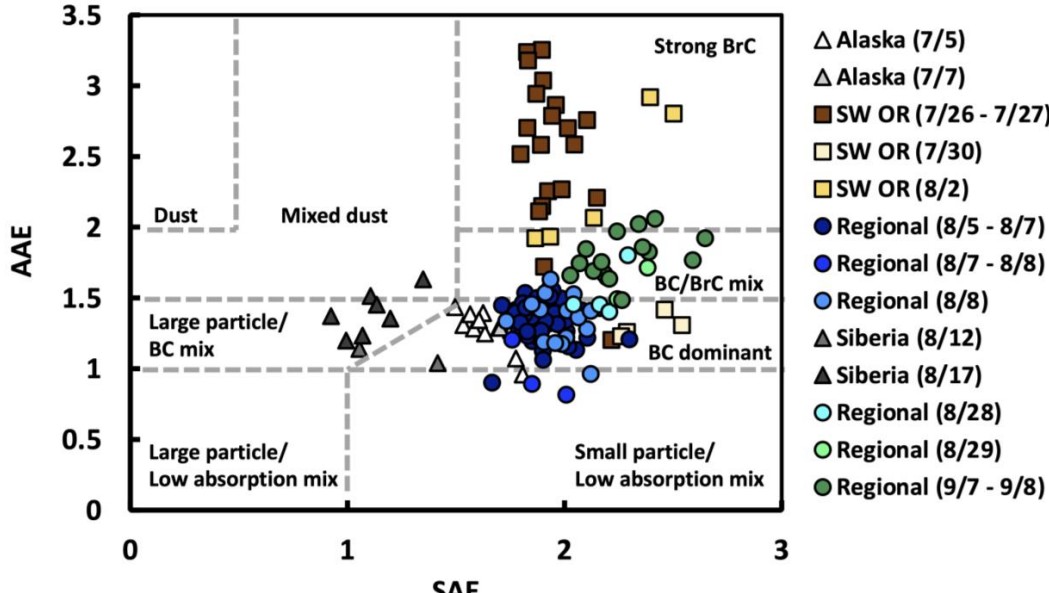

**Figure 3:** 1 hr average SAE and AAE values plotted for all BB events July – September 2019 observed at MBO, with classification categories from Cappa et al. 2016. SAE describes particle size with small value indicating large size and AAE describes particle composition with larger values indicating greater BrC content. BB events are colored by source, with Boreal (Alaska, Siberia) shown as triangles in shades of grey, SW OR shown as squares in shades of brown, and Regional shown as circles in shades of blue to green.



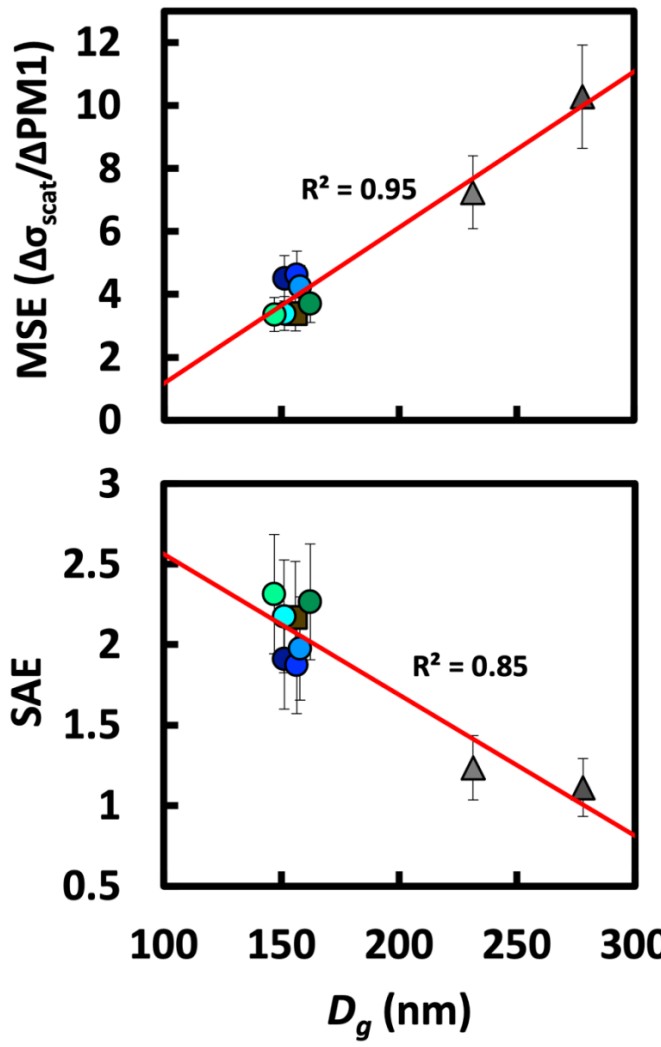


**Figure 4: (a)** Geometric mean diameter of SMPS size distribution ($D_g$) and mass scattering efficiency (MSE) for August – September 2019 BB events observed at MBO, with linear regression and correlation coefficients shown. **(b)** $D_g$ and scattering Angstrom exponent (SAE) for August – September 2019 events observed at MBO, with linear regression and correlation coefficients shown. 2019 BB events are colored by source location in the same manner as **Figure 3**, with regional events as circles in shades of blue to green, SW OR events as squares in shades of brown, Siberia events as triangles in dark gray.






### 3.5 Satellite Identification of Dust Mixed with Smoke in Siberian BB Events

The NASA Dust Score satellite product, which is computed from infrared channels of the NASA Aqua/AIRS satellite (DeSouza-Machado et al., 2010), was overlaid onto MODIS images to examine the presence of dust mixed with smoke emitted and transported from identified source fires. Dust Score is a qualitative representation of the presence of dust in the atmosphere, with values ranging from 400-500 here. The sensor resolution is 45 km with once per day coverage. A

significant Dust Score (>360) was not observed near the origin of Alaskan, SW OR and regional BB events. In contrast, a large geographic area of dust mixed and transported with smoke is observed in time series (8/5, 8/7, 8/10) of Dust Score overlaid on MODIS images of Siberian Boreal Forest fires **(Figure 5)**.

        We used V3.30 aerosol classification products from the Cloud-Aerosol Lidar with

Orthogonal Polarization (CALIOP) instrument on the Cloud-Aerosol Lidar Infrared Pathfinder Satellite Observation (CALIPSO) satellite (Winker et al., 2010) to further examine long-range transport of dust mixed with smoke for Alaskan and Siberian BB events. A time series (8/10, 8/12, 8/17) of the NASA CALIPSO aerosol type satellite cross-sections suggests "dust", "polluted dust", and "smoke" underwent transport in the FT from Siberia to the western USA **(Figure S10)**. A time

series (6/30, 7/2, 7/5) of the NASA CALIPSO aerosol type satellite cross-sections shows "polluted continental/smoke" and "smoke", with comparatively minor contributions from "dust", transported from AK to the western USA **(Figure S11)**. CALIPSO has previously been used in a similar manner to identify the transport of dust with smoke in plumes of Asian origin arriving at MBO in Spring (Fischer et al., 2009). This result suggests that fires in the Siberian Boreal Forest,

as well as other non-arid regions, should be considered as a source of airborne mineral dust.





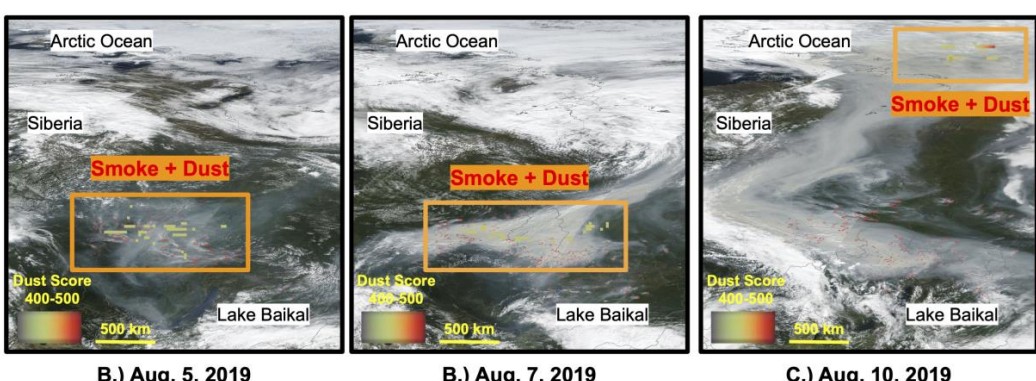

**Figure 5:** (A) August 1, (B) August 5, and (C) August 10, 2019, NASA MODIS true color
satellite images of Siberian Boreal Forest fires. Fire and Thermal Anomalies shown in red dots
and Dust Score values shown in pixels colored according to the embedded scale. Areas of Dust
Score and visible smoke outlined in orange boxes.

### 3.6 Aerosol Size Distribution & Composition

Aerosol number size distribution (30-600 nm) was measured by SMPS for 9 BB events, as
well as interstitial background periods, in August and September 2019 (**Figure 6**). Geometric
diameters ($D_g$) of aerosol size distributions are presented in **Table 1**, with further size distribution
parameters detailed in **Table S2**. The SW OR and regional events exhibited similar unimodal size
distributions, with $D_g$ values between 146 – 162 nm. A prominent "tail" consisting of higher-than-
expected number concentrations of small-diameter particles (30–90 nm) was observed in the size
distribution for events arriving in the BL at MBO similar to results in Laing et al., 2016.
Background periods exhibited a unimodal size distribution with a smaller $D_g$ (101 nm). Siberian
events were characterized by bimodal size distributions, with a larger accumulation mode peak
>200 nm and a lower ultrafine mode peak <100 nm **(Figure 6)**.

The larger accumulation modes of Siberian events are consistent with fine dust particles
mixed with BB aerosol identified in the Dust Score and CALIPSO satellite products. Submicron
dust particles as small as 300 nm have been observed in Africa (Kaaden et al., 2009; Kandler et
al., 2009; Denjean et al., 2016) and Asian dust storms have been found to be correlated with an
increase in 200-500 nm particles (Liang et al., 2013). Furthermore, submicron dust particles in the
size range measured by SMPS in this study are expected to be transported more efficiently than
the coarse dust particles that may have been present in the initial Siberian BB plume. Farley et al.,










2022 measured a larger mass weighted size distribution by SP-AMS for Siberian BB events compared to regional BB events. However, without aerosol chemical composition measurements of dust, which is not measured by SP-AMS, we cannot conclude with confidence that dust was present in the Siberian BB events.


Ultrafine particles are typically not the result of long-range transport, as it would be expected that small particles would grow through coagulation relatively quickly. Therefore, the observed <100 nm modes in long-range transported Siberian BB events are indicative of influence of entrained background air and/or new particle formation (NPF). A year-long study at the Swiss mountaintop observatory Jungfraujoch suggested summer NPF to be triggered by previously entrained precursors or by boundary layer injections on the same day (Tröstl et al., 2016). Nucleation events at Storm Peak, a mountaintop observatory in the Western USA, in spring are most likely associated with sulfuric acid, whereby NPF events in summer are driven by low-volatility organics (Yu and Hallar, 2014). Farley et al., 2022 found Siberian BB to be associated with low volatility oxidized organic aerosol, greater O:C elemental ratio, and greater secondary sulfate mass, the later consistent with observations by Sihto et al., 2006 and Schill et al., 2020. BB smoke transported long-range distances thus likely mixes with other aerosol sources, such as sulfate from marine or other sources, leading to an enhancement in small particles.




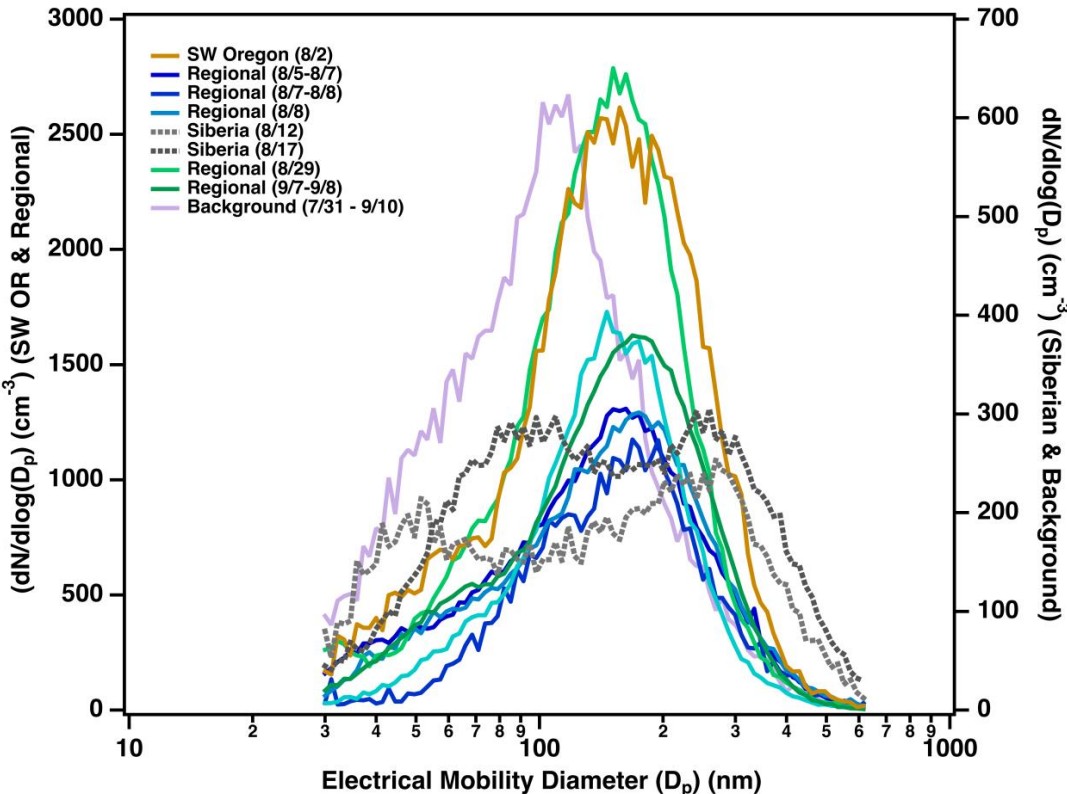

**Figure 6:** Event-averaged aerosol number size distributions in d$N$/dlog($D_p$) measured by SMPS for the SW OR (left axis), Regional (left axis), and Siberian (right axis) BB events identified in August – September 2019 at MBO, as well as background average aerosol number size distribution
(right axis).

## 4. Conclusions

We characterized the physical and optical properties of 13 aged biomass burning events observed at the Mt. Bachelor Observatory in summer 2019. Our conclusions were as follows:

• Low $\Delta PM_1/\Delta CO$ of Siberian BB events are likely the result of enhanced wet deposition in trans-Pacific and trans Arctic FT transport from Siberia to MBO.

• BB events from Siberian and Alaskan Boreal Forest fires with multi-day transport in the FT were associated with lower AAE values indicative of BC dominance with little BrC. Siberian BB events exhibited elevated MAE compared to all other BB events. We propose
two hypotheses for the BC dominance of Boreal Forest BB events:



- o The short half-life of BrC leads to its decays over the course of hours with little remaining in plume after a week or more of transport.
        - o Siberian events with higher MAE originated from hotter, more flaming portions of the fires.
- BB events from a SW Oregon wildfire arriving in the BL with short transport time (<15 h) were associated with the highest AAE indicative of BrC contributions to absorption.
    - The higher $D_g$, lower SAE, and higher MSE of Siberian BB events reflected contributions from large particles, for which we propose two hypotheses:
        - o Enhanced secondary processing during long-range transport leading to particle
growth, which is supported by SP-AMS measurements of enhanced low-volatility organics and sulfate in Siberian BB reported by Farley et al., 2022.
        - o Fine dust transported alongside smoke, which is supported by CALIPSO and Dust Score satellite products of dust mixed with smoke originating from Siberian Boreal Forest fires.


## 5. Data availability

The Mt. Bachelor Observatory 2019 dataset are permanently archived at the University of Washington Research Works site:

https://digital.lib.washington.edu/researchworks/discover?scope=%2F&query=%22mt.+bachelor
+observatory%22&submit=&filtertype_0=title&filter_relational_operator_0=contains&filter_0=data

**Acknowledgements**

The Mt. Bachelor Observatory is supported by the National Science Foundation (grant #AGS-
1447832) and the National Oceanic and Atmospheric Administration (contract #RA-133R-16-SE-0758). Ryan Farley and Qi Zhang acknowledge funding support by NSF (grant # AGS-1829803). The authors gratefully acknowledge the NOAA Air Resources Laboratory (ARL) for the provision of the HYSPLIT transport model used in this publication. The CALIPSO satellite products were supplied from the NASA Langley Research Center.






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
