# Peer review of "Intensive aerosol properties of Boreal and regional biomass burning aerosol at Mt. Bachelor Observatory: Larger and BC-dominant particles transported from Siberian wildfires"

_Atmospheric Chemistry and Physics, 2022_

## Author Response (AR1)

We thank the reviewers for their thoughtful and valuable comments. We also apologize for the long delay in submitting revision. This occurred due to the fact that the primary author switched positions and institutions and was unable to continue working on the manuscript.

We have carefully revised the manuscript according to the comments. Our point-to-point responses can be found below, with reviewer comments repeated in black and author responses in blue. Changes made to the manuscript are in quotation marks.

Responses to Anonymous Referee #2

This manuscript by May et al. presents field measurements of BB aerosols transported to the MBO site, including long-range transported smoke from Alaskan and Siberian boreal forest wildfires and emissions from regional wildfires.

The measurements characterized the physical and optical properties of BB aerosols for plume age from 10 h to 14+ days. The major findings include (1) This work supports the widespread influence of different wildfire emissions on aerosol properties in the western US; (2) The short and long-range transported plume aerosols present different physical and optical properties, which are related to sources and various chemical and physical processes during transport. The paper is well-written and is a valuable contribution to the BB studies. I have some comments detailed below.

Major comment:

1. Section 3.4, Page 12: the discussion of $\Delta PM_1/\Delta CO$

1) Please check the unit of $\Delta PM_1/\Delta CO$. It should be ($\mu g\ m^{-3}\ ppbv^{-1}$)?

Corrected, thank you.

2) Line 285-287: Suggest adding more information, i.e. "The concurrent measurement of decreasing $\Delta OA/\Delta CO$ with increased transport time in Farley et al. (2022) is still because that net OA loss through evaporation and deposition was great than the secondary processing. And OA is the majority component of PM1. Thus, the trend of $\Delta PM_1/\Delta CO$ follows the $\Delta OA/\Delta CO$, which can support the observation of PM1 loss in this study".

Done.

2. Section 3.4, Page 13: the discussion of MAE and BC dominance

Line 318: I prefer not to use "BC enhancement" in this para, it is "the normalized enhancement ratios of BC ($\Delta rBC/\Delta CO$) in Siberian events were identified higher than other cases" (Farley et al., 2022). In Farley et al. (2022), the highest $\Delta rBC/\Delta CO$ ratios were measured during both Siberia and Oregon events, suggesting that these events had more influence from flaming fires. It is still not explained why the Siberian events exhibited a higher average MAE (0.60 $m^2\ g^{-1}$) than the SW OR events (0.30 $m^2\ g^{-1}$). Suggest checking BC fraction in PM1.

We have replaced the term "BC enhancement" with "the normalized enhancement ratios of BC (ΔrBC/ΔCO)". The plume data were reanalyzed and Table 1 has been updated. More explanations are given on the observed higher MAE for Siberian events.  The revised text is pasted below on this page.

The mean MAE in the 8/12 Siberian event (0.48 m2 g−1) is actually close to most other events, the mean MAE in the 8/17 Siberian event (0.72 m2 g−1) is much higher, which leads to the higher average MAE of Siberia events. What caused this difference between the 8/12 and 8/17 Siberia events?

The plume data were reanalyzed and associated values have been updated.

The discussions indicate that Siberia events in this study had more influence from flaming fires, thus having a high ΔBC/ΔCO and MAE. I agree with this, more flaming fires do generate plumes with enhanced BC emissions (ΔBC/ΔCO). However, in this study, the ΔBC/ΔCO ratios are not only related to the source fire conditions but also the transport processes, i.e., Siberia events are suggested to experience wet deposition. It's needed to mention that although the Siberia events experienced strong wet deposition, the ΔBC/ΔCO ratios are still higher than other events. A study (https://doi.org/10.1029/2020GL088858) observed enhanced fraction of BC after vertical transport from the surface to the top of the boundary layer due to the lower removal efficiency of BC than the non-BC materials and the evaporation of other non-BC materials, which may be also related the transport processes.

We have revised the corresponding texts according to the review comments. They revised texts now read:

"Examination of intensive aerosol absorption properties suggests that the Siberian BB aerosol were generally more absorptive and BC-dominant than other BB events observed. The average MAE of the Siberian BB events (0.40 $m^2$ $g^{-1}$) is similar to the regional events (0.43 $m^2$ $g^{-1}$) but higher than the AK and SW OR events (0.29 and 0.24 $m^2$ $g^{-1}$, respectively).  The greater MAE in Siberian events are consistent with higher $BC/PM_1$ ratios in Siberian events than other cases identified by Farley et al. (2022) via the concurrent SP-AMS measurements at MBO.  A possible reason for the high mass fractions of BC in Siberian aerosols was wet deposition and enhanced oxidative aging during long-range transport (Farley et al., 2022), which remove BC at a lower rate than the non-BC materials (Liu et al., 2020). Another possible reason for the higher MAE in Siberian events is greater influence from flaming fires. Indeed, the Siberian events, despite wet deposition and prolonged atmospheric aging, showed higher normalized enhancement ratios of BC (ΔBC/ΔCO) than other cases (Farley et al., 2022)"

3. Line 343: A recent laboratory study also found the imaginary part of BrC could be half decayed in a few hours, in line with the loss of its absorptivity after transport (https://doi.org/10.1021/acs.est.0c07569).

The laboratory work by Cappa et al., 2020 (hhttps://doi.org/10.5194/acp-20-8511-2020) and field observation from Wu et al., 2021 (https://doi.org/10.5194/acp-21-9417-2021) suggest that the evolution of AAE and BrC absorptivity with photochemical aging is dependent on

the fire burn conditions and initial emission particle properties. There is an initial enhancement stage of AAE and BrC absorptivity followed by the decrease with longer aging times for more flaming fires, while more smouldering fires are suggested to experience a net decrease upon aging. The short-range transported (10-15 h) SW OR events with the highest AAE may experience the initial enhancement stage or decrease during this short-range transport period depending on the fire condition. Suggest adding more clarification here.

We have made revisions according to these comments. The revised texts now read:

"A recent laboratory study also found the imaginary part of BrC could be half decayed in a few hours, in line with the loss of its absorptivity after transport (Liu et al., 2021). Furthermore, the laboratory work by Cappa et al. (2020) and field observation from Wu et al. (2021) suggest that the evolution of AAE and BrC absorptivity with photochemical aging is dependent on the burn conditions and the initial properties of emitted particles. Aerosol from more flaming fires show an initial enhancement of AAE and BrC absorptivity followed by a gradual decrease at longer aging times, while more smoldering aerosol are suggested to experience net decrease upon aging. Thus, the highest AAE of the SW OR events could also be related to burning conditions and the initial enhancement of AAE during the short-range transport."

Line 393-395: "Notably, a portion of the 8/17 Siberian event exhibited a combination of elevated AAE and low SAE that is typically indicative of dust aerosols with enhanced absorption at short wavelengths." Siberia events did exhibit lower SAE than other events due to larger size mode particles. However, the AAE values in Siberia events were not elevated compared to other events. What does the "elevated AAE" mean? The AAE values in Siberia events were close to another long-range transport event (Alaska event). I don't think the AAE can be evidence of the dust aerosols mixing with plumes.'

We have made revisions according to these comments. The revised texts now read:

"While secondary aerosol growth and/or coagulation during transport present plausible explanations for larger particles, we also consider the unexpected contribution of dust mixed with smoke. SAE has previously been used by Zhang and Jaffe 2016 to identify larger particle sizes in long-range transport of industrial pollution from Asia, which was suggested to be due to mixing with mineral dust sources. In addition, prior measurements in spring have identified an association of dust mixed with smoke in long-range transported plumes from Asia with lower SAE at MBO (Fischer et al., 2010). Elevated AAE alongside low SAE (Figure 3) can be indicative of dust aerosols with enhanced absorption at short wavelengths (Bergstrom et al., 2007; Titos et al., 2017). However, the distribution and average AAE of the Siberian BB events were similar to the Alaskan BB events."

4. Page 22: For the aerosol size distribution in Siberia events, I agree that the observed <100 nm modes may be indicative of the influence of entrained background air and/or new particle formation. However, in Siberia events, the suggested wet deposition during transport would remove larger-size BB aerosols and would also result in a smaller size mode under 100 nm. Examples from Taylor et al., 2014 (https://doi.org/10.5194/acp-14-13755-2014) can support this. This is also related to the conclusion on Page 24.

This has been added at line 501.

Specific comments:

Line 143: need a full name for "PSAP".

Done.

Line 171: Please check the correction of OPC PM1 measurements in the supplementary, not found.

There is not much to add to what is in the methods section.  During the thermo-denuder measurements, the OPC data were adjusted to match the SMPS mass concentrations.  This is now added to the methods section.

Line 198: How do you get the BB event criteria of $\sigma$scat > 20 Mm$^{-1}$ and CO > 110 ppbv?

This is based on significant enhancements above the background levels and our prior work (Wigder et al 2013)

Line 214: In Figure1, need to add annotation (the blue, green and red lines in $\sigma$abs and $\sigma$scat plots represent blue, green and red channels respectively).

Done.

Line 220-221: Two BB events (7/5-7/7), rather than the (7/5-7/6) in the manuscript. The $\Delta$WV of Alaska events is -1.52 in Table 1, not consistent with -1.53 here. Some values in Table 1 are not consistent with the manuscript, please check them.

Done

Line 239: Which prior observations? If indicating previous work, please add the reference.

Wigder et al 2013.  Reference has been added

Page 11, Table 1: It would be good to add notes explaining why some of the data are missing in Table 1 (not measured or not good quality data?).

Table 1 have been updated.

Line 313: "in the / during Siberia BB event" repeat word? Please re-phase

Done.

Line 314: the result *"of"* increased aqueous-phase cloud processing during transport

Done.

Line 330: ARCTAS-A aircraft measurements in Alaska *"reported"* a much larger BC/CO ratio.

Done.

Line 337: What is "PNW" BB event? BB event from where?

This sentence has been revised to:

"Analysis of 10 years of aerosol properties at MBO by (Zhang and Jaffe, 2017) found Asian long-range transport wildfires with lower AAE (1.45 ± 0.02 and 1.54 ± 0.39) than regional BB events (1.81 ± 0.59)."

Line 369/370: Please explain what is "Dpm"?
Done.

Responses to Anonymous Referee #3

This paper summarizes the results of measurements made at Mt Bachelor over a roughly 2-month period. A variety of instruments were used to determine dry particle optical properties (absorption and scattering, each at 3 wavelengths), and particle number size distributions for particles in the diameter range of 30 to 600nm.  Over this period, 13 biomass burning events were identified.  The likely regions of wildfires producing the events were determined by back trajectory analysis, and properties of the events compared between sources (or ages of smoke).  A possible dust event mixed with smoke is also identified.  The paper is very well written, and care is taken in producing the data and evaluating uncertainties.  However, sitting at a fixed site and observing periods of smoke blow by puts limitations on interpreting the smoke events. The authors spend considerable time trying to explain possible causes for the observed differences between events and those of other studies, which in most cases is largely speculation. I recommend the authors attempt a more robust analysis that would include more support (eg, statistical or theoretical calculations) for their many hypotheses to explain the differences.  Overall, the methods nor the results are highly novel, but they do provide very useful data on smoke plumes of various approximate ages in the real atmosphere; possible causes for the variability discussed throughout the manuscript needs more attention.

It's not really clear what the reviewer meant by "statistical or theoretical calculations".  In any case, this is not a theoretical study, but reports on observations.

Specific Comments:

Line 49-50, this definition of BrC (the aerosol overall AAE>2) is highly measurement specific, it reflects the fact that most methods, such as those deployed in this study, cannot detect low levels of BrC (ie, instruments that cannot isolate BrC, have very limited spectral resolution, and have lowest measurement wavelengths at rather high values

where BrC contributions are not strong, such as in this study with lowest wavelength 450nm). The authors should clarify how this affects what they specify as BrC. For example, for a given wavelength, say 450 nm and 350 nm, for various AAEs, what fraction of the light absorption coefficient is due to BrC, and include an uncertainty in this ratio that considers the effect of the fit (ie, the assumption of a power law and the specific two wavelengths used, and the variability in pure BC AAE). Finally, does this method of determining contributions of BrC depend on the assumption that BrC falls on a continuum from weak to strongly absorbing BrC, see (Saleh, R. (2020), From Measurements to Models: Toward Accurate Representation of Brown Carbon in Climate Calculations, Current Poll. Rep., 6, 90-104).

I think you are proposing a different study. In this work, our goals are to assess different types of biomass plumes and their impact on aerosol properties. This includes the light absorption properties, as measured by the absorption at 467 and 660. We do not attempt to determine the absorption fraction due to brown carbon.

The point here is that there are subtleties and limitations in determining BrC based simply on AAEs – which should discuss.

We acknowledge that there are other different "types" of brown carbon and different methods that can determine BrC with higher sensitivity and specificity. To address this comment, we have added this sentence at line 53.

"Saleh (2020) describe four classes of brown carbon, with widely varying optical properties and point out that the BrC properties may depend on the measurement method".

Does the SAE and AAE calculated depend on the specific 2 wavelengths selected for the calculation. Eg, how much would they vary if the other two pairs were used in the calculation?

The plots below show how the SAE and AAE values vary depending on the wavelength for the 13 biomass burning plumes identified in our study. While the values do change, the relative relationships do not. Not surprisingly, using a wider wavelength range for AAE generates larger values. For the SAE using 700 nm in the calculation generates greater noise due to lower scattering values at the longer wavelengths. No changes to the manuscript are needed here.

[Figure]

[Figure]

How much mass is missed by particles smaller than the OPC lower size limit? This could be determined by estimating the mass from the SMPS.

This is a good question.  We have added the following text at line 169:

"The OPC measures the particle size distribution for particles with aerodynamic diameters from 0.25 to 32 μm in 31 size bins.   We note that this size ranges misses a significant fraction of particle mass for smaller particles.   For the biomass burning plumes observed in this study, approximately 42% of the $PM_1$ mass is accounted for by particles with diameters <250 μm as determined by the SMPS data. The OPC $PM_{2.5}$ mass concentrations are determined via the manufacturer's calibration against standard filter methods (Grimm and Eatough, 2009).  The OPC is a USA EPA Federal Equivalent Method for measuring $PM_{2.5}$ mass concentrations."

Line 178, at what wavelength was the single scattering albedo determined at?

SSA was calculated at 528 nm.   This is now added to the methods (line 180).  Scattering at 550 nm was adjusted to 528 nm using a power law relationship, as shown in equation 1.

Lines 275 to 295 is largely speculation since the NEMR's at the sources are not known and NEMRs from only 2 other studies are used as a contrast.  I suggest this analysis be changed by providing an in-depth discussion of various dPM1/dCO recorded close to wildfires from as many data sets as can be found, and then compare that and the variability to the data from this study.  The authors could then more quantitatively assess if they are observing differences in processes or if it is hard to say bases on variability in emissions that have been recorded.  I would point out that dOA/dCO could likely be used as a surrogate for dPM1/dCO since most of the mass is OA, as the authors state in the Introduction. This will likely significantly expand the published data that can be used given all the recent aircraft missions studying wildfires.  Over-interpretation of the data is common throughout this paper.  Broad statistical comparisons, such as shown in Fig 2 are more convincing than speculation.  Can a Fig similar to Fig 2 be made for dPM/dCO?

The NEMR's of wildfire emissions are influenced by many factors, such as fuel type, burn conditions, and atmospheric aging.   Figure 2 shows intensive aerosol properties.

Because of the significant CO background (80-100 ppb), we feel the RMA regression of PM$_1$ vs CO (shown in Table 2) is a better way to show this data.

Lines 329 to 333, the interpretation here is that differences in observed BC/CO can be used to infer BC/CO at the sources (ie, flaming vs smoldering). Again, this is speculation. It may be one possible reason but there could be others, such as differential loss of BC relative to CO during smoke transport (contrast the typical lifetimes of BC and CO if precipitation is encountered).

Agree. The revised texts now read:

"Examination of intensive aerosol absorption properties suggests that the Siberian BB aerosol were generally more absorptive and BC-dominant than other BB events observed. The average MAE of the Siberian BB events (0.40 m$^2$ g$^{-1}$) is similar to the regional events (0.43 m$^2$ g$^{-1}$) but higher than the AK and SW OR events (0.29 and 0.24 m$^2$ g$^{-1}$, respectively).  The greater MAE in Siberian events are consistent with higher BC/PM$_1$ ratios in Siberian events than other cases identified by Farley et al. (2022) via the concurrent SP-AMS measurements at MBO.  A possible reason for the high mass fractions of BC in Siberian aerosols was wet deposition and enhanced oxidative aging during long-range transport (Farley et al., 2022), which remove BC at a lower rate than the non-BC materials (Liu et al., 2020). Another possible reason for the higher MAE in Siberian events is greater influence from flaming fires. Indeed, the Siberian events, despite wet deposition and prolonged atmospheric aging, showed higher normalized enhancement ratios of BC (ΔBC/ΔCO) than other cases (Farley et al., 2022).  We also note that the low AAE values in the Siberian plumes is consistent with bleaching during atmospheric transport (Andersson et al 2019).

Ling 346-347. The authors might want to note a contrasting paper, Dasari, S., A. Andersson, S. Bikkina, H. Holmstrand, K. Budhavant, S. Satheesh, E. Asmi, J. Kesti, J. Backman, A. Salam, D. Bisht, S. Tiwari, and Z. H. O. Gustafsson (2019), Photochemical degradation affects the light absorption of water-soluble brown carbon in the South Asian outflow, Sci. Advances, 5, eaau8066.

This has been noted (line 346).

Fig 4 and associated text. The correlations and slopes shown depend on two points out of 8 or so. Can one infer from this that there is a general relationship here?  Provide statistical proof.

Yes, this really depend on the two Siberian plumes.   Nonetheless, the plots are statistically significant with p-values < 0.05.

Line 459-460. This last line sounds like the authors are making a generalization based solely on two observed events.  Is that reasonable?  It is rather unfortunate that no filters were collected that could be used to measure dust (Ca2+) and smoke tracers (K+, along with the measured BC) in the same event.  This would provide proof that the plumes were

indeed mixtures of smoke and mineral dust, the analysis presented is only suggestive. (This was noted on line 490, but I would point this out earlier in this discussion.

This sentence was revised in response to this comment. The texts now read:

"However, due to lack of direct measurement of dust in this study, we cannot conclude with confidence that dust was present in the Siberian BB events. Nevertheless, the CALIPSO result suggests that fires in the Siberian Boreal Forest, as well as other non-arid regions, should be considered as a source of airborne mineral dust."

Section 3.6 title and text within, define exactly what type of size distribution is being discussed, ie number distributions.

Done.

Lines 480 and on, would not a calculated volume distribution provide better evidence for a possible dust influence in the SMPS measurement size range. Ie, one could compare the shape of volume distributions for the non-dust and speculated dust events.

We have revised the sentence, which now reads:

"The larger accumulation modes of Siberian events suggest contributions from fine dust particles identified in the Dust Score and CALIPSO satellite products."

Lines 491 to the end is mainly speculation. Why not estimate the lifetime of an UF particle based on the measured number distributions to support this discussion.

Agree. We have deleted this discussion.

In conclusions, bullet 2. What is meant by little BrC. (See earlier comment). Given the very insensitive method the authors used to determine BrC, this is rather a subjective statement. Same applies to the term little remaining…. The point is, using the method for determining BrC in this paper, exactly what fraction of the light absorption at a given wavelength (the authors may choose) is due to BC vs BrC and include an uncertainty. Maybe use this instead of the term little.

We have modified this statement to read: "The aerosols during these events have lower AAE values compared to regional bb events." (line 535).